# L-C4: Language-Based Video Colorization for Creative and Consistent Colors

## Abstract

Automatic video colorization is inherently an ill-posed problem because each monochrome frame has multiple optional color candidates. Previous exemplar-based video colorization methods restrict the user's imagination due to the elaborate retrieval process. Alternatively, conditional image colorization methods combined with post-processing algorithms still struggle to maintain temporal consistency. To address these issues, we present Language-based video Colorization for Creative and Consistent Colors (L-C4) to guide the colorization process using user-provided language descriptions. Our model is built upon a pre-trained cross-modality generative model, leveraging its comprehensive language understanding and robust color representation abilities. We introduce the cross-modality pre-fusion module to generate instance-aware text embeddings, enabling the application of creative colors. Additionally, we propose temporally deformable attention to prevent flickering or color shifts, and cross-clip fusion to maintain long-term color consistency. Extensive experimental results demonstrate that L-C4 outperforms relevant methods, achieving semantically accurate colors, unrestricted creative correspondence, and temporally robust consistency.

## 1 Introduction

Video colorization aims to convert monochrome videos into plausible colorful versions and is widely used in film restoration, artistic creation, and advertising. The objective of video colorization is to assign semantically accurate and visually pleasing colors while ensuring temporal consistency to maintain a smooth visual experience without flickering or color shifts. To achieve this goal effectively, researchers have explored various approaches, *e.g.*, automatic video colorization methods (Lei & Chen, 2019; Thasarathan et al., 2019; Liu et al., 2024) that infer colors from monochrome semantic cues, exemplar-based video colorization methods (Zhang et al., 2019; Iizuka & Simo-Serra, 2019; Yang et al., 2024c) that transfer provided exemplar colors to monochrome ones, and conditional image colorization methods (Chang et al., 2023b; Huang et al., 2022; Chang et al., 2023a) combined with post-processing algorithms (Lai et al., 2018; Lei et al., 2020; 2023) to remove flickering artifacts during the colorization process.

Although these advancements demonstrate great potential for video colorization in diverse applications, they still have several limitations due to their inherent task settings: *(i)* **Ambiguous color assignment.** Automatic video colorization methods face the ill-posed nature of the task, struggling when an instance has multiple optional color candidates. This can lead to colorization results that may not accurately meet users' expectations (Fig. 1 first row). *(ii)* **Limited creative imagination.** Exemplar-based video colorization requires users to provide reference images. Since the process of retrieving appropriate references can be elaborate, and the retrieved images are generally collected in the wild, these methods may restrict the ability to assign creative colors to instances based on users' imagination (Fig. 1 second row). *(iii)* **Vulnerable color consistency.** Although conditional image colorization methods combined with post-processing algorithms enable colorizing videos with more user-friendly interactive conditions (*e.g.*, language descriptions), existing models still struggle to maintain temporal consistency when handling significant variations in object positions and deformations across frames (Fig. 1 third row).

In this paper, we propose an innovative language-based video colorization framework to understand user-provided language descriptions without requiring post-processing, effectively addressing

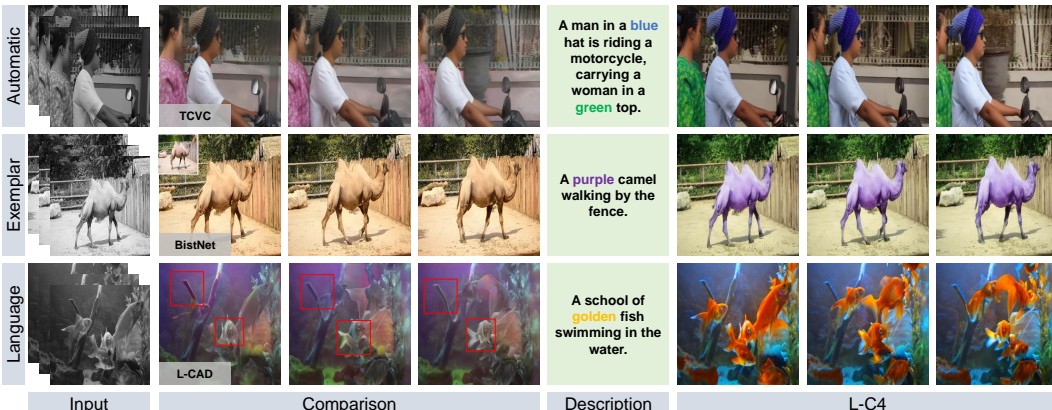

Figure 1: Advantages of our language-based video colorization framework compared to relevant colorization methods (Liu et al., 2024; Yang et al., 2024c; Chang et al., 2023a): **First row:** Automatic methods cannot specify the color of each garment, whereas the language-based method explicitly establishes this correspondence to meet users' expectations. **Second row:** Exemplar-based method cannot colorize the camel purple due to the difficulty of finding appropriate references, whereas the language-based method allows users to apply creative colors freely. **Third row:** Image colorization method combined with post-processing algorithms struggles to maintain color consistency when the fish swims rapidly, whereas the language-based method demonstrates greater robustness.

the aforementioned issues: *(i)* **Semantically accurate colors.** Our model is built upon the cross-modality generative model (Rombach et al., 2022), leveraging its comprehensive language understanding and robust color representation abilities to assign semantically accurate and visually pleasing colors that meet users' expectations. *(ii)* **Unrestricted creative correspondence.** We present the cross-modality pre-fusion module to generate instance-aware text embeddings. This module can correctly assign specified colors to corresponding instances within video clips, enabling the application of creative colors. *(iii)* **Temporally robust consistency.** To ensure robust inter-frame consistency, we propose temporally deformable attention, which lifts color priors from images to videos and prevents flickering or color shifts by effectively capturing each instance and maintaining similar feature representation. Additionally, we introduce the cross-clip fusion to extend the temporal interaction scope, maintaining long-term color consistency when colorizing long videos.

We name our approach **L-C4**, the **L**anguage-based video **C**olorization for **C**reative and **C**onsistent **C**olors. Compared to previous works, L-C4 offers the following advantages: *(i)* Due to the flexible nature of language descriptions, users can explicitly colorize monochrome videos according to their instructions (Fig. 1 first row). *(ii)* Using instance-aware text embeddings liberates the color-object correspondence from real-world constraints, enabling users to assign colors to each instance with creative colors (Fig. 1 second row). *(iii)* The language-based video framework constructs spatial-temporal interactions, demonstrating robustness in maintaining color consistency with instances that are moving or deforming (Fig. 1 third row). We summarize our contributions as follows:

- We propose a language-based video colorization model that enables user-friendly interactions and produces temporally consistent and visually creative colorization results.
- We present the temporally deformable attention and the cross-modality pre-fusion module to ensure inter-frame color consistency and instance-aware colorization, respectively.
- We introduce the cross-clip fusion that achieves long video colorization while maintaining color consistency during the inference phase.

## 2 RELATED WORK

### 2.1 VIDEO COLORIZATION

The video colorization task requires models to render semantically accurate and visually pleasing colors for targeted monochrome videos while maintaining temporal consistency across frames. Au-

tomatic video colorization methods are trained to infer colors from the semantic cues presented in monochrome video frames, without human intervention. Although researchers explore various approaches to improve colorization performance, *e.g.*, self-regularization (Lei & Chen, 2019), generative adversarial learning (Zhao et al., 2023), and optical flow estimation (Liu et al., 2024), the problem remains inherently ill-posed. To guide the colorization process, exemplar-based approaches (Wan et al., 2022; Zhang et al., 2019) gain significant attention, which leverage reference images or user-selected frames to construct implicit correspondences with exemplar features. DeepRemaster (Iizuka & Simo-Serra, 2019) introduces a fully 3D convolutional framework to extract temporal-spatial video features. BiSTNet (Yang et al., 2024c) addresses issues of object occlusion and color bleeding by incorporating bidirectional feature fusion and semantic segmentation priors. While these methods can produce plausible results, they heavily rely on the relevance and quality of the chosen exemplars, limiting their applicability in diverse scenarios.

## 2.2 LANGUAGE-BASED IMAGE COLORIZATION

Language-based image colorization aims to colorize monochrome images according to user-provided language descriptions, providing a flexible and user-friendly interaction approach. Manjunatha *et al*. (Manjunatha et al., 2018) is the first to introduce this task, designing the feature-wise affine transformations (Perez et al., 2018) to inject language descriptions into the colorization process. Similarly, LBIE (Chen et al., 2018) develops a recurrent attentive model to spatially fuse image and text features. To ensure the colorization results accurately meet users' expectations, L-CoDe (Weng et al., 2022) and L-CoDer (Chang et al., 2022) utilize additional annotated correspondences between object and color words, addressing the problem of color-object coupling. Towards instance awareness, L-CoIns (Chang et al., 2023b) and L-CAD (Chang et al., 2023a) introduce a grouping mechanism and a sampling strategy, respectively. Recently, researchers pay more attention to integrating multiple conditions within a unified framework (Huang et al., 2022; Liang et al., 2024), which further lowers the barrier for color assignment. When it comes to colorizing monochrome videos, combining image colorization with post-processing algorithm (Lai et al., 2018; Lei et al., 2020; 2023) might seem reasonable but often leads to performance degradation in practical applications. Therefore, to guide the video colorization with language descriptions effectively, it is necessary to design a model tailored to this task.

## 2.3 VIDEO DIFFUSION MODEL

Since diffusion models demonstrate dominance in image generation (Rombach et al., 2022; Zhang et al., 2023), researchers explore approaches to lift pre-trained image generative priors to video. To overcome the challenges of modeling temporal dependencies, Ho *et al*. (Ho et al., 2022b) extend denoising networks with 3D convolutions and train the model from scratch based on large-scale video datasets. However, this approach significantly increases computational resource demands. To address this, ControlVideo (Zhang et al., 2024) proposes a training-free strategy with inter-frame global attention. Other methods (Esser et al., 2023; He et al., 2022; Ho et al., 2022a; Xing et al., 2024) focus on introducing temporal modules (*e.g.*, temporal convolution (Carreira & Zisserman, 2017) and temporal attention (Bertasius et al., 2021)) into existing image diffusion models (Rombach et al., 2022; Saharia et al., 2022). To preserve the generative priors of the pre-trained model, they only fine-tune the added corresponding module. Recently, researchers turn their attention to generating long videos, exploring techniques like temporal co-denoising (Wang et al., 2023a) and noise rescheduling (Qiu et al., 2024). Despite extensive research in video generation, existing methods still face challenges in maintaining the long-term color consistency of instances.

## 3 METHOD

In this section, we begin by providing an overview of our framework (in Sec. 3.1). Subsequently, we describe details of the temporally deformable attention that ensures inter-frame color consistency (in Sec. 3.2) and cross-modality pre-fusion module that generates instance-aware text embedding (in Sec. 3.3). Following this, we introduce the cross-clip fusion (in Sec. 3.4), designed to maintain long-term color consistency when colorizing long videos. Finally, we elaborate on details of network training (in Sec. 3.5).

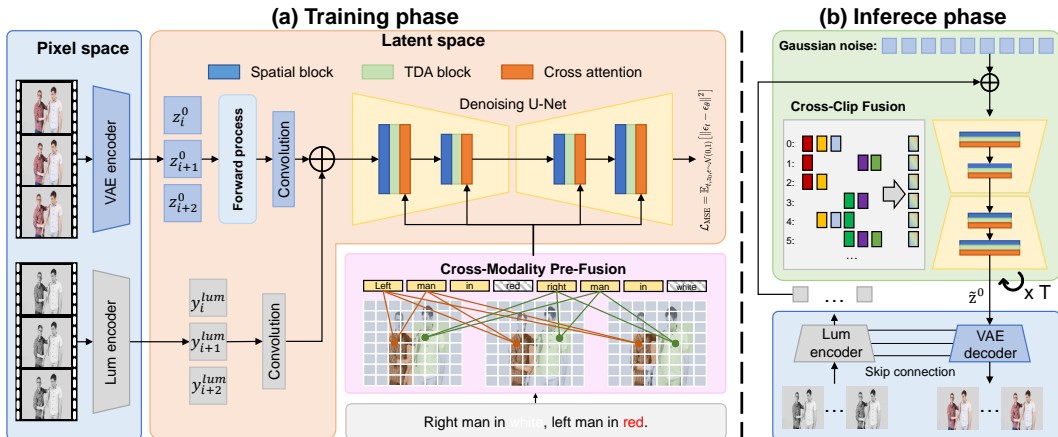

Figure 2: The pipeline of L-C4. (a) During the training phase, video frames are projected into the latent space with a VAE encoder, and noise is subsequently added. The monochrome features $y^{\mathrm{lum}}$ extracted by the luminance (lum) encoder are added to the noised latent codes to align the global structure with the monochrome frames. We equip the denoising U-Net with the Temporally Deformable Attention (TDA) block, ensuring robust inter-frame color consistency. We present the Cross-Modality Pre-Fusion (CMPF) module to generate instance-aware text embeddings, enabling the application of creative colors for specified instances. (b) During the inference phase, we introduce the Cross-Clip Fusion (CCF) to maintain long-term color consistency when colorizing long videos. When decoding the predicted latent code $\tilde{z}^0$, multi-scale monochrome features from the luminance encoder are added into the corresponding scales of the VAE decoder through skip connections to preserve local details.

## 3.1 OVERVIEW

We illustrate the framework of L-C4 in Fig. 2. As a latent diffusion model, L-C4 performs the forward and backward processes in the latent space. Specifically, given an $N$ frames video clip $X = \{x_i\}_{i=1}^N$, L-C4 adopts a pre-trained VAE encoder $\mathcal{E}$ to project each frame $x_i$ into latent code $z_i^0 = \mathcal{E}(x_i)$, and a pre-trained VAE decoder $\mathcal{D}$ to reconstruct the video frames as $\tilde{x}_i = \mathcal{D}(z_i^0)$. The pre-trained weights of the VAE are obtained from SD1.5 (Rombach et al., 2022).

To preserve the global structure and local details of the monochrome video $X^{\mathrm{lum}} = \{x_i^{\mathrm{lum}}\}_{i=1}^N$, we additionally introduce a luminance encoder $\mathcal{E}^{\mathrm{lum}}$ that shares the structure with $\mathcal{E}$ to extract monochrome features as $y_i^{\mathrm{lum}} = \mathcal{E}^{\mathrm{lum}}(x_i^{\mathrm{lum}})$. This brings two advantages: *(i)* The extracted features could align the global structure between the noised latent code and the monochrome frames. Specifically, they are added to the latent code after a convolution layer. *(ii)* The multi-scale monochrome features could preserve local details during the latent code decoding. Specifically, we add them into the corresponding scales in the VAE decoder using skip connections. In practice, the pre-trained weights of $\mathcal{E}^{\mathrm{lum}}$ are obtained from L-CAD (Chang et al., 2023a).

To maintain the temporally consistent representations across frames, we equip the denoising U-Net with Temporally Deformable Attention (TDA) (in Sec. 3.2). The TDA is inserted between the spatial blocks (*i.e.*, residual blocks and spatial self-attention blocks) and cross-attention blocks to capture the inter-frame dependency of hidden features. To achieve the instance-aware colorization, we introduce Cross-Modality Pre-Fusion (CMPF) to generate the instance-aware text embeddings by improving the semantic representation of noun concepts (in Sec. 3.3). These embeddings are injected into the denoising U-Net via cross-attention blocks. To colorize long videos, we propose Cross-Clip Fusion (CCF) to maintain long-term color consistency while reducing the computational consumption (in Sec. 3.4). CCF is only performed during the inference phase.

## 3.2 INTER-FRAME COLOR CONSISTENCY

To ensure temporally consistent representations across frames and avoid flickering or color shifts, we propose the Temporally Deformable Attention (TDA), illustrated as the green block in the denoising

U-Net in Fig. 2 (a). Previous temporal attention (Blattmann et al., 2023; Esser et al., 2023; He et al., 2022) extracts context at fixed locations, while the global inter-frame attention (Zhang et al., 2024) may introduce irrelevant regions. Considering the assumption that colors in a video do not change dramatically over short periods, we compress the time dimension and adopt a deformable receptive field to capture dynamic features of instances in video colorization. Our proposed TDA could effectively capture each instance across frames while keeping their feature representations similar, regardless of significant variations in positions and deformation. As shown in Fig. 3, the TDA is calculated as the following steps:

*(i)* **Reference points sampling.** Denote the hidden features as $h \in \mathbb{R}^{N^{\mathrm{f}} \times H \times W \times C}$, where $N^{\mathrm{f}}$ represents the number of frames, $H$ and $W$ are the height and width, respectively, and $C$ means the number of channels. To effectively reduce the consumption of computation, we uniformly sample reference points in the video to construct the sampled coordinate sequence $p \in \mathbb{R}^{\hat{H} \times \hat{W} \times \hat{N}^{\mathrm{f}} \times 3}$, where $N^{\mathrm{rs}}$ and $N^{\mathrm{rt}}$ are separately the sampling rates for resolution and time. As a result, the sampling resolutions are significantly reduced to $[\hat{H}, \hat{W}, \hat{N}^{\mathrm{f}}] = [H/N^{\mathrm{rs}}, W/N^{\mathrm{rs}}, N^{\mathrm{f}}/N^{\mathrm{rt}}]$. These reference points represent a compressed spatiotemporal space of the video clip.

*(ii)* **Offset estimation.** To calculate the relevant context for the frame sequence, we further estimate the offset for each reference point. Specifically, we project the frame sequence $h$ via the linear projection $W_{\mathrm{o}}$, following a lightweight 3D convolution network $\delta_{\mathrm{offset}}(\cdot)$ to generate the offsets. To further ensure that the estimated context covers the overall semantics, we confine the range of offsets into $[-1, 1]$ as $\Delta p = \tanh(\delta_{\mathrm{offset}}(hW_{\mathrm{o}}))$.

*(iii)* **Context extraction.** With the estimated offsets, we could extract relevant context with deformed points and transform them into keys $\tilde{k}_i$ and values $\tilde{v}_i$ via corresponding projection matrices:

$$\tilde{h} = \phi(h; p + \alpha \Delta p), \quad \tilde{k}_i = \tilde{h} W_{\mathrm{k}}^i, \quad \tilde{v}_i = \tilde{h} W_{\mathrm{v}}^i, \tag{1}$$

where $\alpha$ is a hyperparameter to control the range of candidate context, $i \in \{1, \dots, N^{\mathrm{h}}\}$ is the index of attention heads, and $\phi(\cdot; \cdot)$ is the function that weights estimated deformed points, formulated as:

$$\phi(h; (p_x, p_y, p_t)) = \sum_{(r_x, r_y, r_t)} g(p_x, r_x) g(p_y, r_y) g(p_t, r_t) h[r_t, r_y, r_x], \tag{2}$$

where $g(a, b) = \max(0, 1 - |a - b|)$ and $(r_x, r_y, r_t)$ indexes locations of hidden features $h$. Finally, TDA performs in a multi-head attention manner, formulated as:

$$\hat{h} = \mathrm{Concat}_{i \in \{1, \dots, H\}} \Big( \mathrm{Softmax}\big((q_i \tilde{k}_i^\top)/\sqrt{d}\big) \tilde{v}_i \Big) W_{\mathrm{h}}, \quad q_i = h W_{\mathrm{q}}^i, \tag{3}$$

where $W_{\mathrm{q}}$ and $W_{\mathrm{h}}$ are learnable matrices and $d$ is the number of channels.

## 3.3 INSTANCE-AWARE TEXT EMBEDDING

Previous language-based image colorization model (Chang et al., 2023a) achieves instance awareness by introducing extra instance segmentation annotations and modifying the sampling strategy of the denoising process. However, this faces challenges in tracking instances that may be occluded or move in and out of frames when colorizing monochrome videos. To avoid introducing external requirements while correctly assigning colors to corresponding instances in video clips, we propose the Cross-Modality Pre-Fusion (CMPF) module as shown in the pink section of Fig. 2 (a). This module generates instance-aware text embeddings by improving the semantic representation of noun concepts, ensuring creative colorization results.

Specifically, the CMPF module comprises $L$ fusion blocks, each consisting of a Multi-head Self-Attention (MSA), a Masked Cross-Attention (MCA), and a Feed-Forward Network (FFN). Denote CLIP embeddings (Radford et al., 2021) of user-provided language descriptions at the $l$-th block as $y_l^{\mathrm{tex}} \in \mathbb{R}^{N^{\mathrm{t}} \times C}$, where $l \in \{1, \dots, L\}$ and $N^{\mathrm{t}}$ is the length of language descriptions. We first adopt an MSA to learn the inter-word modification relationships between embeddings as:

$$\bar{y}_{l-1}^{\mathrm{tex}} = \mathrm{LN}\big(\mathrm{MSA}(y_{l-1}^{\mathrm{tex}}) + y_{l-1}^{\mathrm{tex}}\big), \tag{4}$$

where LN is the layer normalization. Next, using the monochrome features $y^{\mathrm{lum}}$ as the visual prompt, we extract compressed features based on the aforementioned TDA as $\hat{y}^{\mathrm{lum}} = \mathrm{TDA}(y^{\mathrm{lum}})$.

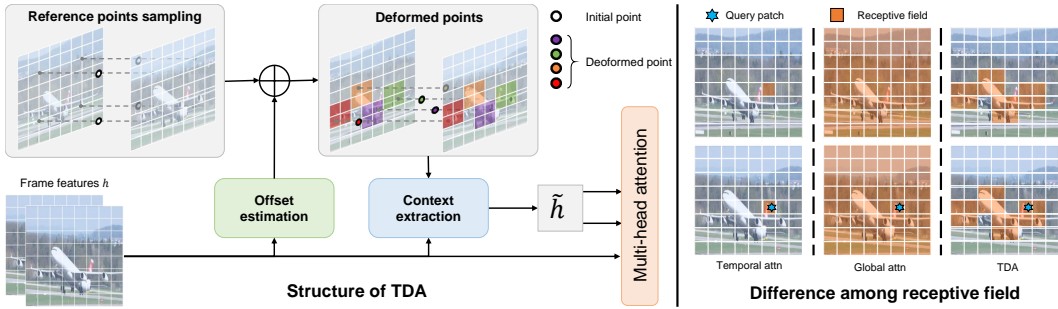

Figure 3: Illustration of TDA's structure and the different receptive fields. **Left:** We uniformly sample reference points and estimate offsets for each point to calculate deformed points. After that, we could extract relevant context with multi-head attention. **Right:** Previous temporal attention only extracts context at fixed spatial locations across frames, struggling to find relevant context when objects move or deform (*e.g.*, the plane's tail wing). The global attention may introduce features from irrelevant regions (*e.g.*, calculating all features) and bring excessive computational consumption. Our proposed TDA can accurately capture relevant context across frames via the estimated deformed points, effectively addressing the aforementioned limitations.

We perform the MCA to establish the correspondences between embeddings $\bar{y}_{l-1}^{\text{tex}}$ and the compressed monochrome features $\hat{y}^{\text{lum}}$. Note that we apply a mask $M$ to exclude the color-related words (*e.g.*, "red" and "white"), allowing the model to focus on the understanding of noun concepts:

$$\tilde{y}_{l-1}^{\text{tex}} = \text{LN}\big(\text{MCA}(\bar{y}_{l-1}^{\text{tex}}, \hat{y}^{\text{lum}}, M) + \bar{y}_{l-1}^{\text{tex}}\big). \tag{5}$$

Following that, the FFN integrates the combined features as:

$$y_l^{\text{tex}} = \text{LN}\big(\text{FFN}(\tilde{y}_{l-1}^{\text{tex}}) + \tilde{y}_{l-1}^{\text{tex}}\big). \tag{6}$$

After $L$ times iteration, CMPF generates the instance-aware text embeddings $y_L^{\text{tex}}$. Before injecting these embeddings into the denoising network, we apply a learnable linear layer $W_{\text{e}}$, initialized to zero, and add them with the initial CLIP embeddings $y_0^{\text{tex}}$ to provide an efficient initialization:

$$y^{\text{tex}} = W_{\text{e}}(y_L^{\text{tex}}) + y_0^{\text{tex}}. \tag{7}$$

### 3.4 LONG-TERM CONSISTENT INFERENCE

Existing methods struggle to colorize long videos consisting of hundreds of frames due to computational resource limitations. A trivial approach is to independently colorize multiple non-overlapping video clips and then stitch them together. Intuitively, this strategy fails to ensure color consistency across video clips. Some colorization methods (Lei & Chen, 2019; Yang et al., 2024c) adopt autoregressive strategies by referring to previously colorized frames. However, this strategy is prone to error accumulation. While some methods (Liu et al., 2024; Wan et al., 2022) improve temporal consistency based on optical flow, they heavily rely on the precision of flow estimation. Inspired by recent diffusion-based fusion mechanisms (Wang et al., 2023a), we introduce the Cross-Clip Fusion (CCF) to effectively extend the temporal interaction scope and maintain long-term color consistency when colorizing long videos.

Instead of using a sliding window to capture local context, we use a skip window to capture long-term color dependency. Specifically, we select $N^{\text{f}}$ frame video clips with different time intervals $d \in \{1, 2, 4, \ldots, N^{\text{d}}\}$, where $N^{\text{d}}$ is the largest power of 2 that is less than the total number of video frames. To achieve a smooth temporal experience, we fuse these cross-clip features for each frame. This process is visualized in Fig. 2 (b), where frames within the same clip are represented by patches of the same color. Given that frames closer to the central frame of the clip tend to include more representative features, we further implement the distance-based weighted fusion to alleviate inconsistencies at clip boundaries, instead of direct averaging:

$$\hat{f}_i = \sum_j \frac{\left(N^{\text{f}}/2 - |i - c_{i,j}|\right) f_{i,j}}{\sum_k \left(N^{\text{f}}/2 - |i - c_{i,k}|\right)}, \tag{8}$$

Table 1: Quantitative results on two evaluation benchmarks. Throughout this paper, $\uparrow$ ($\downarrow$) means higher (lower) is better. Best performances are highlighted in **bold**.

| Method | DAVIS30 | | | | | | Videvo20 | | | | | |
|---|---|---|---|---|---|---|---|---|---|---|---|---|
| | Color. $\uparrow$ | PSNR $\uparrow$ | SSIM $\uparrow$ | LPIPS $\downarrow$ | FVD $\downarrow$ | CDC $\downarrow$ | Color. $\uparrow$ | PSNR $\uparrow$ | SSIM $\uparrow$ | LPIPS $\downarrow$ | FVD $\downarrow$ | CDC $\downarrow$ |
| AutoColor | 23.69 | 23.20 | 0.919 | 0.239 | 962.72 | 3.671 | 23.20 | 23.73 | 0.923 | 0.258 | 1381.01 | 1.960 |
| VCGAN | 14.26 | 24.26 | 0.927 | 0.235 | 1367.31 | 5.208 | 14.53 | 23.94 | 0.920 | 0.223 | 1351.53 | 3.753 |
| TCVC | 19.71 | 25.04 | 0.922 | 0.224 | 1143.53 | 3.855 | 19.74 | 24.84 | 0.929 | 0.219 | 975.41 | 1.956 |
| DeepExemplar | 23.84 | 24.51 | 0.905 | 0.230 | 1175.74 | 4.229 | 25.11 | 23.21 | 0.916 | 0.236 | 794.11 | 2.013 |
| DeepRemaster | 18.49 | 24.33 | 0.913 | 0.225 | 1073.95 | 5.400 | 20.93 | 23.40 | 0.904 | 0.208 | 883.05 | 3.437 |
| BiSTNet | 27.00 | 24.88 | 0.928 | 0.214 | 977.15 | 3.946 | 25.30 | 24.43 | 0.924 | 0.217 | 749.85 | 1.974 |
| L-CoIns | 17.06 | 23.04 | 0.871 | 0.236 | 1474.44 | 3.946 | 16.46 | 23.44 | 0.872 | 0.260 | 1387.66 | 2.027 |
| UniColor | 19.30 | 22.74 | 0.851 | 0.238 | 1245.69 | 4.569 | 18.71 | 23.04 | 0.857 | 0.246 | 1198.54 | 2.884 |
| L-CAD | 19.80 | 22.70 | 0.886 | 0.212 | 1347.26 | 4.407 | 21.98 | 23.30 | 0.881 | 0.221 | 962.72 | 2.945 |
| *W/o* TDA | 29.01 | 25.24 | 0.915 | 0.219 | 761.85 | 3.646 | 29.15 | 25.06 | 0.927 | 0.229 | 468.31 | 1.775 |
| *W/o* CMPF | 28.72 | 24.84 | 0.923 | 0.227 | 724.63 | 3.673 | 28.72 | 24.82 | 0.921 | 0.224 | 476.52 | 1.658 |
| *W/o* CCF | 29.19 | 25.23 | 0.927 | 0.216 | 735.76 | 3.593 | 29.51 | 24.74 | 0.935 | 0.204 | 496.81 | 2.335 |
| Ours (L-C4) | **29.33** | **25.69** | **0.933** | **0.209** | **654.32** | **3.114** | **32.59** | **25.17** | **0.939** | **0.198** | **420.59** | **1.572** |

where $|\cdot|$ denotes absolute value function, $c_{i,j}$ represents the index of the center frame in the $j$-th video clip that contains the $i$-th video frame, and $f_{i,j}$ denotes the feature of the $i$-th frame extracted by the TDA block in the $j$-th video clip.

### 3.5 Learning and Implementation Details

We adopt the two-stage training strategy for our model. In the first stage, we train the denoising U-Net $\epsilon_\theta$ under the guidance of the CLIP embeddings (Radford et al., 2021) of language descriptions. In the second stage, we freeze the denoising U-Net and train the CMPF module to optimize the instance-aware embeddings. We apply the mean squared error (MSE) loss to optimize the model in both stages:

$$\mathcal{L}_{\text{MSE}} = \mathbb{E}_{t,z_0,\epsilon_t \sim \mathcal{N}(0,1)}\left[\|\epsilon_t - \epsilon_\theta(z^t, t, y^{\text{tex}}, y^{\text{lum}})\|^2\right]. \tag{9}$$

Our model is trained on the subset of the InternVid dataset (Wang et al., 2023b), which comprises 100K text-video pairs, and all samples have an aesthetic score above 5.5. We train the first stage over 20 epochs with a batch size of 2, spending approximately 100 hours. We use the AdamW optimizer with a learning rate of $1 \times 10^{-5}$ and momentum parameters $\beta_1 = 0.99$ and $\beta_2 = 0.999$. In the second stage, we use the same settings and train 10 epochs. In this paper, we set the clip length for training $N^{\text{f}} = 8$ and hyperparameter $\lambda = 1 \times 10^{-4}$. All experiments are conducted on 6 NVIDIA V100 graphics cards.

## 4 Experiment

We perform comparisons and ablation studies on two widely used benchmarks: the DAVIS (Perazzi et al., 2016) and the Videvo (Lai et al., 2018) datasets. Adhering to the evaluation protocol of existing video colorization methods (Liu et al., 2024; Zhao et al., 2023; Yang et al., 2024c), we conduct evaluation experiments on the DAVIS validation set (30 videos) and the Videvo test set (20 videos). Following the most relevant language-based colorization methods (Chang et al., 2023b; Huang et al., 2022; Chang et al., 2023a), we select a resolution of $256 \times 256$ to ensure a fair comparison. During training, we leverage the language descriptions provided by InternVid Wang et al. (2023b), which employs InstructBLIP (Dai et al., 2023) to generate descriptions for every 20-frame interval and then integrate these descriptions into a single coherent caption for each video using GPT-4 (Achiam et al., 2023). During evaluation, we recruit human volunteers to annotate language descriptions to evaluate the practical applicability.

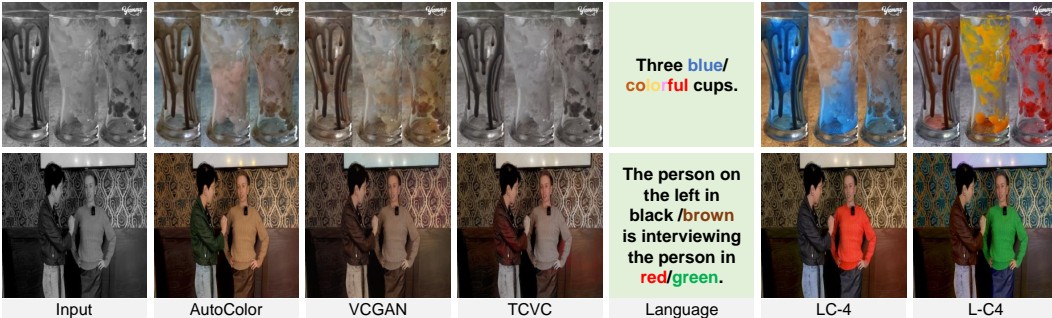

Figure 4: Visual quality comparison with automatic video colorization methods (Lei & Chen, 2019; Zhao et al., 2023; Liu et al., 2024).

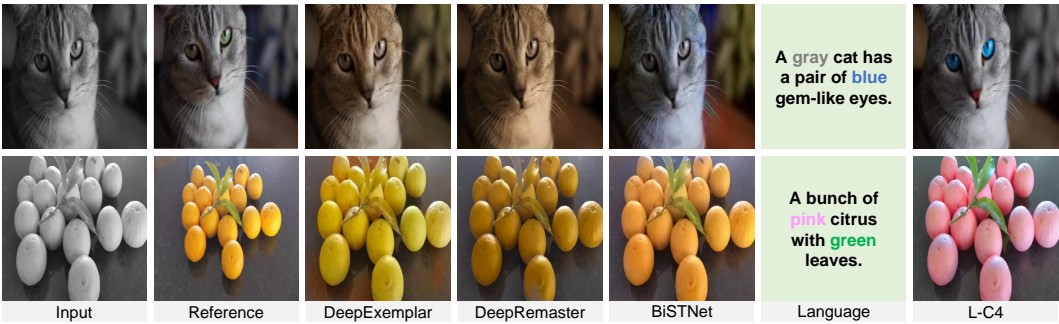

Figure 5: Visual quality comparison with exemplar-based video colorization methods (Zhang et al., 2019; Iizuka & Simo-Serra, 2019; Yang et al., 2024c).

## 4.1 COMPARISON WITH STATE-OF-THE-ART METHODS

We make comparisons with 3 automatic video colorization methods (*i.e.*, AutoColor (Lei & Chen, 2019), TCVC (Liu et al., 2024), and VCGAN (Zhao et al., 2023)), 3 exemplar-based colorization methods (*i.e.*, DeepExemplar (Zhang et al., 2019), DeepRemaster (Iizuka & Simo-Serra, 2019), and BiSTNet (Yang et al., 2024c)), and 3 language-based image colorization methods (*i.e.*, L-CoIns (Chang et al., 2023b), UniColor (Huang et al., 2022) and L-CAD (Chang et al., 2023a)) combined with the post-processing algorithm (Lei et al., 2023) to demonstrate the effectiveness of our L-C4 in the color consistency, instance awareness, and creative colorization. Note that only language-based colorization methods Chang et al. (2023b); Huang et al. (2022); Chang et al. (2023a) share the same task setting as our approach. Other video colorization methods are presented to highlight the advantages of our task setting. All comparison experiments are conducted using the officially released code for each method.

**Evaluation metrics.** At the frame level, we employ the colorfulness (Color.) score (Hasler & Suesstrunk, 2003) to assess the vividness of colors in the colorization results. Additionally, we report PSNR (Huynh-Thu & Ghanbari, 2008), SSIM (Wang et al., 2004), and LPIPS (Zhang et al., 2018) metrics to evaluate the perceptual difference between the colorized frames and the ground truth. At the video level, we utilize the Fréchet Video Score (FVD) (Unterthiner et al., 2019) to quantitatively assess the perceptual realism of the whole colorized videos, which calculates the feature distribution similarity between the colorization results and the ground truth. We further measure the temporal consistency using the Color Distribution Consistency (CDC) index, reported with a scale of 1000 for better readability (Liu et al., 2024).

**Quantitative comparisons.** As shown in Tab. 1, our method achieves the best scores across all evaluation metrics. The best FVD score demonstrates that the overall quality of our colorization surpasses other methods. With the CPFM that generates instance-aware text embeddings, our method ensures accurate colorization results for specified instances, resulting in higher PSNR, SSIM, and

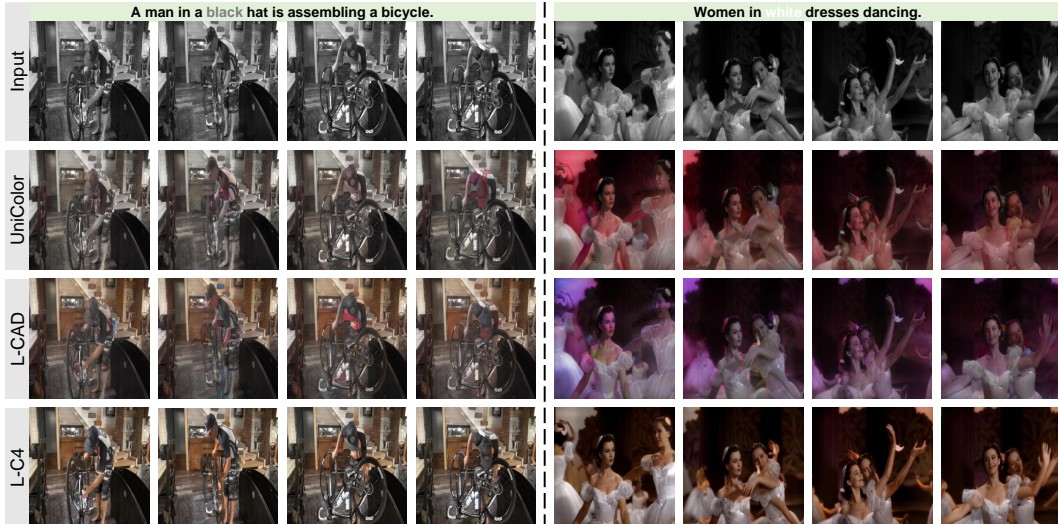

Figure 6: Visual quality comparison with language-based image colorization methods (Huang et al., 2022; Chang et al., 2023a) combined with post-processing algorithms (Lei et al., 2023).

LPIPS scores. Leveraging the robust color representation of the diffusion model, our method achieves more vivid colorization results with a higher colorfulness score. Additionally, equipped with our proposed TDA and CCF, our colorization results present more robust temporal consistency, achieving the best CDC score.

**Qualitative comparisons.** As shown in Fig. 4, L-C4 presents semantically accurate colors. On the top, with the prompt "blue/colorful", the cups are vividly colorized. On the bottom, users explicitly assign the red/green color to the right person to meet their expectations. As shown in Fig. 5, L-C4 achieves unrestricted creative correspondence. On the top, the oranges are colorized pink, a color hardly observed in nature. On the bottom, users can freely assign the cat's appearance, eliminating the need for corresponding exemplars. As shown in Fig. 6, L-C4 demonstrates temporally robust consistency. On the left, the color of the man's hat remains consistently black across frames despite movement. On the right, the color of the dancing woman stays consistent.

### 4.2 USER STUDY

We conduct user studies to evaluate the subjective perception of human observers: *(i)* **Spatial Color Assignment (SCA) experiment:** This experiment assesses whether our method assigns semantically accurate colors more effectively than relevant comparison methods. Participants are shown a random monochrome frame from the video along with 7 colorization results and asked to select the one that presents the best visual quality. *(ii)* **Temporal Color Consistency (TCC) experiment:** This experiment determines whether our method provides more robust color consistency across frames compared to relevant methods. Participants are shown a monochrome video along with 7 colorization results, and instructed to select the one that offers the smoothest visual experience without flickering or color shifts. For each experiment, we randomly select 10 samples from the testing videos of each dataset and let 25 volunteers make their choices independently on the Amazon Mechanical Turk (AMT). As shown in Tab. 2, L-C4 achieves the highest scores in both experiments.

### 4.3 ABLATION STUDY

We create three baselines to study the impact of our proposed modules. The evaluation scores and colorization results of each ablation study are presented in Tab. 1 and Fig. 7, respectively.

***W/o* Temporally Deformable Attention (TDA).** We replace the temporally deformable attention with the temporal attention that extracts context at fixed locations across frames. As a result, this variant struggles to maintain color consistency during instance movement and deformation (*e.g.*, the color of the slate on the wall changes in the first row of Fig. 7, left).

Table 2: User study results. The proposed method produces the highest scores in both experiments.

| Experiment | Dataset | VCGAN | TCVC | DeepRemaster | BiSTNet | UniColor | L-CAD | Ours |
|---|---|---|---|---|---|---|---|---|
| SCA | DAVIS30 | 7.2% | 9.2% | 12.4% | 16.8% | 10.4% | 12.8% | **31.2%** |
| | Videvo20 | 5.2% | 8.8% | 11.6% | 13.2% | 12.4% | 16.0% | **32.8%** |
| TCC | DAVIS30 | 10.4% | 11.6% | 11.2% | 12.4 % | 6.4% | 10.8% | **37.2%** |
| | Videvo20 | 16.4% | 18.0% | 7.6% | 12.8% | 7.2% | 8.8% | **29.2%** |

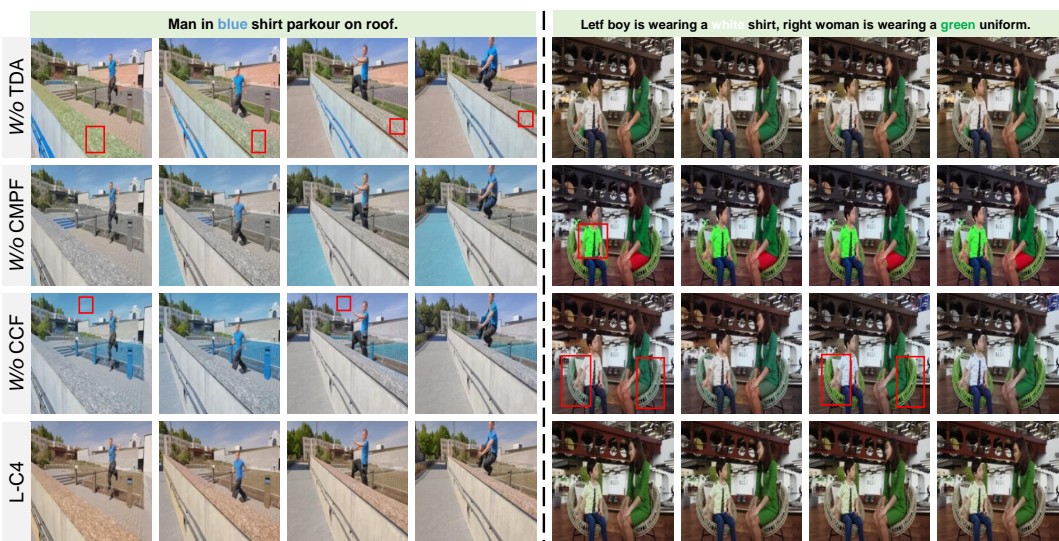

Figure 7: Ablation study results. When our proposed modules are disabled, the results exhibit vulnerable temporal color consistency and limited instance awareness.

***W/o* Cross-Modal Pre-Fusion (CMPF).** We use the text encoder of CLIP (Radford et al., 2021) to encode language descriptions, discarding our proposed cross-modal pre-fusion module. This leads to suboptimal performance in correctly assigning colors to corresponding instances based on users' specific requests (*e.g.*, the boy's clothes turn green in the second row of Fig. 7, right).

***W/o* Cross-Clip Fusion (CCF).** We remove the cross-clip fusion and perform clip-by-clip inference when colorizing long videos. Consequently, this ablation fails to maintain color consistency across video clips, and instances not described tend to be colorized inconsistently (*e.g.*, the sky changes from blue to purple and the chairs shift from gray to green in the third row of Fig. 7, left and right).

## 5 CONCLUSION

In this paper, we propose **L-C4**, an innovative framework for **L**anguage-based video **C**olorization for **C**reative and **C**onsistent **C**olors. Compared to existing colorization methods that suffer from ambiguous instance correspondence, vulnerable color consistency, and limited creative imaginations, L-C4 understands user-provided language descriptions without requiring post-processing and effectively addresses the aforementioned issues. To assign semantically accurate and visually pleasing colors that meet users' expectations, we leverage the generative priors of a pre-trained cross-modality generative model. To correctly assign specified colors to corresponding instances and enable the application of creative colors, we present the cross-modality pre-fusion module to generate instance-aware text embeddings. To ensure robust inter-frame consistency and maintain long-term color consistency when colorizing long videos, we propose temporally deformable attention and cross-clip fusion. Extensive experiments demonstrate the effectiveness of L-C4, achieving the best scores across six qualitative evaluation metrics on two widely used datasets.

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

# A   APPENDIX

## A.1   FAILURE CASES

Despite the various modules and strategies we have designed, our L-C4 still has difficulty distinguishing fine-grained colors. As shown in Fig. 8, our model cannot accurately colorize the car with "Klein blue" or "dark blue". We will continue to explore precise color control in future work.

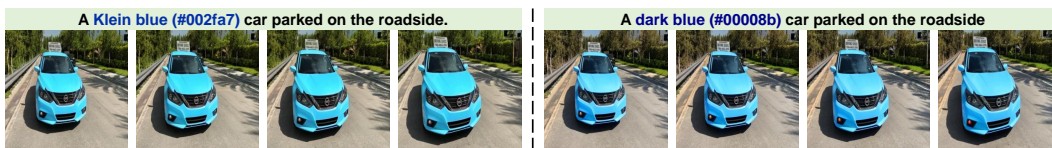

Figure 8: Failure cases in assigning fine-grained colors.

## A.2   ADDITIONAL CHALLENGING DATASET

We conduct an additional experiment to evaluate comparison methods on the randomly selected 30 videos from a subset of VRIPT (Yang et al., 2024a), where all samples have aesthetic scores greater than 5. These videos do not overlap with the training data for any of the comparison methods. Quantitative comparisons are presented in Tab. 3. The results show that our method demonstrates superiority over the compared methods.

Table 3: Additional quantitative results on the VRIPT dataset.

| Method | VRIPT | | | | | |
| --- | --- | --- | --- | --- | --- | --- |
| | Color. ↑ | PSNR ↑ | SSIM ↑ | LPIPS ↓ | FVD ↓ | CDC ↓ |
| AutoColor (Lei & Chen, 2019) | 20.77 | 22.48 | 0.903 | 0.274 | 2461.63 | 5.845 |
| VCGAN (Zhao et al., 2023) | 18.10 | 22.00 | 0.907 | 0.234 | 1885.02 | 5.256 |
| TCVC (Liu et al., 2024) | 22.86 | 23.48 | 0.918 | 0.204 | 1250.74 | 3.737 |
| DeepExemplar (Zhang et al., 2019) | 30.42 | 24.76 | 0.932 | 0.140 | 1040.08 | 4.050 |
| DeepRemaster (Iizuka & Simo-Serra, 2019) | 25.27 | 23.72 | 0.918 | 0.160 | 845.98 | 3.437 |
| BiSTNet (Yang et al., 2024c) | 26.98 | 24.47 | 0.929 | 0.139 | 824.65 | 3.876 |
| L-CoIns (Chang et al., 2023b) | 22.64 | 21.74 | 0.893 | 0.207 | 2389.62 | 4.649 |
| UniColor (Huang et al., 2022) | 24.26 | 21.17 | 0.875 | 0.205 | 2084.76 | 3.761 |
| L-CAD (Chang et al., 2023a) | 25.29 | 23.94 | 0.904 | 0.183 | 1863.65 | 3.761 |
| Ours (L-C4) | **35.25** | **25.23** | **0.947** | **0.131** | **695.11** | **2.345** |

## A.3   ADDITIONAL ABLATION RESULTS

We provide the following fine-grained quantitative ablation results to demonstrate the impact of our proposed modules, as presented in Tab. 4 (top):

**3D convolution.** Replacing the temporally deformable attention with standard 3D convolutions.

**Removing mask.** Removing the mask that excludes the color-related words in the MCA of CMPF.

**Sliding window.** Using a sliding window instead of a skip window to capture context in CCF.

**Direct averaging.** Use direct averaging instead of distance-based weighting to fuse clips in CCF.

**Temporal L-CAD.** Equipping L-CAD with temporal attention to create an intuitive baseline.

We further adjust the hyperparameters to show their impacts on L-C4, as shown in Tab. 4 (bottom):

**Hyperparameter $\alpha$.** Adjusting the range of candidate context used in context extraction of TDA.

**Hyperparameter $N^{\mathrm{f}}$.** Adjusting the number of clip frames used in the skip window of CCF.

Table 4: Additional quantitative ablation results.

| Method | DAVIS30 | | | | | | Videvo20 | | | | | |
|---|---|---|---|---|---|---|---|---|---|---|---|---|
| | Color. ↑ | PSNR ↑ | SSIM ↑ | LPIPS ↓ | FVD ↓ | CDC ↓ | Color. ↑ | PSNR ↑ | SSIM ↑ | LPIPS ↓ | FVD ↓ | CDC ↓ |
| 3D convolution | 28.45 | 25.42 | 0.924 | 0.211 | 774.61 | 3.893 | 28.99 | 25.09 | 0.930 | 0.234 | 472.53 | 2.130 |
| Removing Mask | 29.13 | 25.21 | 0.922 | 0.217 | 716.42 | 3.124 | 29.52 | 25.13 | 0.937 | 0.209 | 445.15 | 1.754 |
| Sliding window | 28.54 | 25.45 | 0.931 | 0.215 | 678.35 | 3.487 | 29.36 | 25.71 | 0.931 | 0.225 | 432.63 | 1.624 |
| Direct averaging | 29.09 | 25.13 | 0.927 | 0.209 | 695.42 | 3.593 | 29.51 | 25.58 | 0.938 | 0.205 | 464.38 | 1.939 |
| Temporal L-CAD | 29.32 | 24.99 | 0.914 | 0.239 | 779.34 | 3.859 | 29.52 | 25.06 | 0.918 | 0.239 | 552.12 | 2.047 |
| $\alpha$=2 | 29.12 | 25.26 | 0.929 | 0.224 | 694.34 | 3.135 | 31.35 | 24.92 | 0.915 | 0.223 | 487.75 | 1.624 |
| $\alpha$=8 | 28.32 | 25.62 | 0.917 | 0.216 | 662.63 | 3.536 | 31.56 | 25.22 | 0.924 | 0.236 | 449.41 | 1.834 |
| $N^f$=4 | 28.45 | 25.50 | 0.921 | 0.225 | 667.36 | 3.354 | 30.45 | 25.42 | 0.925 | 0.215 | 460.98 | 1.858 |
| $N^f$=12 | 28.24 | 25.62 | 0.921 | 0.217 | 673.73 | 3.145 | 31.45 | 25.43 | 0.932 | 0.225 | 482.67 | 2.053 |
| Ours (L-C4) | **29.33** | **25.69** | **0.933** | **0.209** | **654.32** | **3.114** | **32.59** | **25.65** | **0.939** | **0.198** | **420.59** | **1.572** |

## A.4 ADDITIONAL VISUAL QUALITY COMPARISON

In the main paper, we only show visual quality comparison with representative methods due to the space limit. As shown in Fig. 9, we show additional comparison results with automatic video colorization methods (*e.g.*, AutoColor (Lei & Chen, 2019), VCGAN (Zhao et al., 2023), and TCVC (Liu et al., 2024)), exemplar-based video colorization methods (*e.g.*, DeepExemplar (Zhang et al., 2019), DeepRemaster (Iizuka & Simo-Serra, 2019), and BiSTNet (Yang et al., 2024c)), and language-based image colorization methods (*i.e.*, L-CoIns (Chang et al., 2023b), UniColor (Huang et al., 2022) and L-CAD (Chang et al., 2023a)) combined with the post-processing algorithm (Lei et al., 2023) to demonstrate our advantages.

In addition, although ColorDiffuser (Liu et al., 2023) also presents a language-based video colorization model, they do not release their code or provide a detailed technical description for reproduction. Therefore, we could only compare our results with the frames presented in their paper. Our method predicts colors with higher saturation (Fig. 10, top) and produces more accurate semantic colors (Fig. 10, bottom left) as well as better preservation of local structures (Fig. 10, bottom right).

As illustrated in Sec. 4.3, we create three baselines to evaluate the efficacy of our proposed TDA, CMPF, and CCF modules. We provide an additional visual quality comparison in Fig. 11.

## A.5 ADDITIONAL APPLICATION RESULTS

We employ our method to colorize old black-and-white films using language descriptions. As the colorization results presented in Fig. 12, L-C4 demonstrates strong generalization capabilities, making the restoration process accessible to non-expert users. Additionally, we present colorization results of L-C4 applied to real-world videos in Fig. 13.

## A.6 ADDITIONAL DISCUSSIONS

In this subsection, we provide additional discussions about the potential concerns: *(i)* We retrain our model at a 512 resolution and present results in Fig. 14, which demonstrate improved handling of fine details compared to the 256 resolution models. *(ii)* We evaluate the robustness by showing the box plots of the quantitative metrics in Fig. 15 (left). To further evaluate the success rate, we additionally conduct a user study. As shown in Fig. 15 (right), over 87% of volunteers rate the colorization quality as "Acceptable" or higher. *(iii)* We perform comprehensive comparisons with the state-of-the-art automatic image colorization method (DDColor (Kang et al., 2023)) and the concurrent exemplar-based video colorization methods (ColorMNet (Yang et al., 2024b)) in Fig. 16 and Tab. 5. *(iv)* We illustrate the impact of CMPF on the attention maps in Fig. 17, demonstrating that L-C4 with CMPF can more accurately identify the corresponding instances compared to the baseline without CMPF. *(v)* We present colorization results using the intricate prompts and show the result in Fig. 18, which demonstrates that L-C4 can effectively handle complex cases. *(vi)* We report inference times in Tab. 6, expand the user study scale in Tab. 7, and list all training datasets used for comparison methods in Tab. 8.

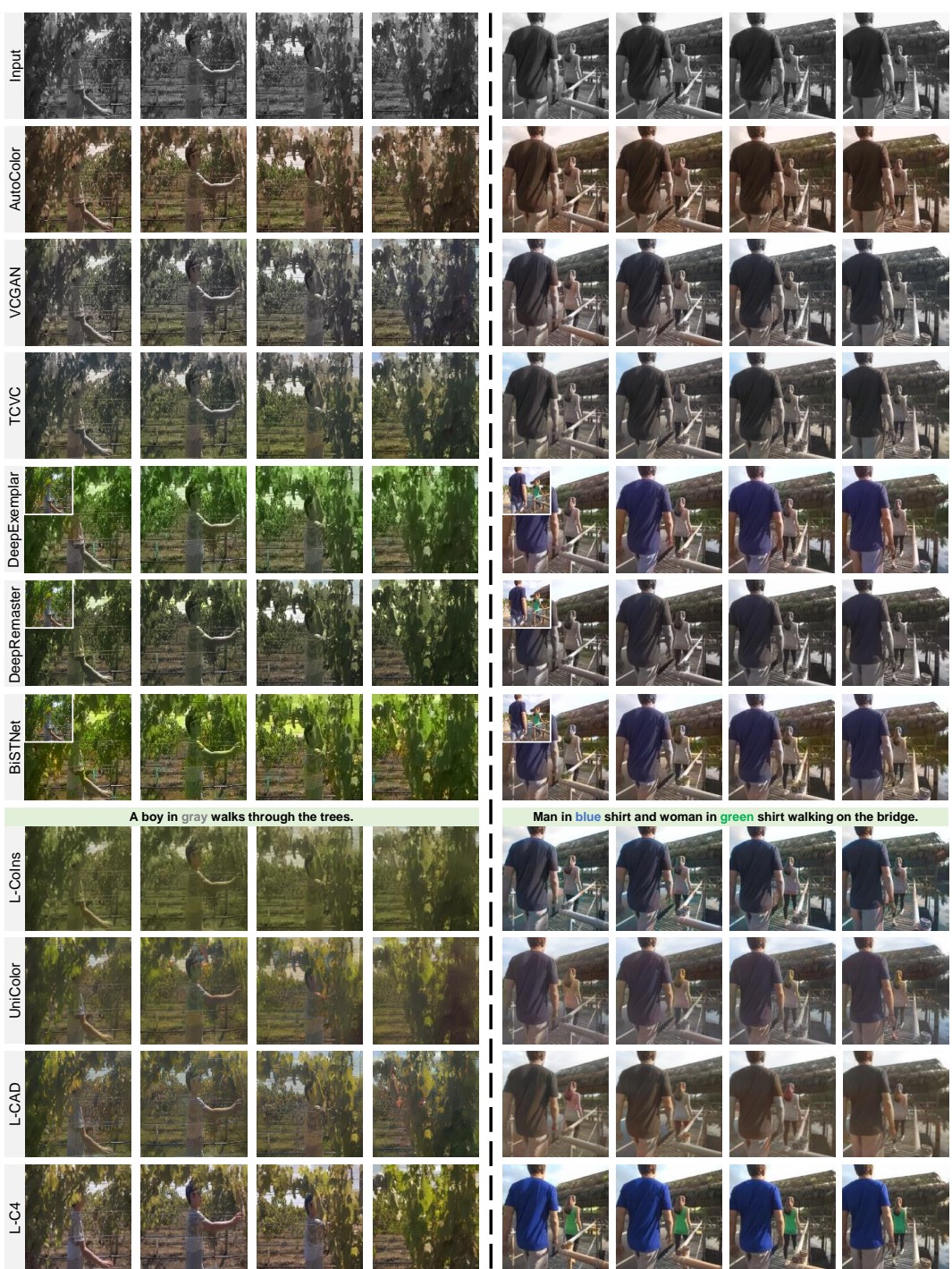

Figure 9: Additional visual comparison results with relevant video colorization methods.

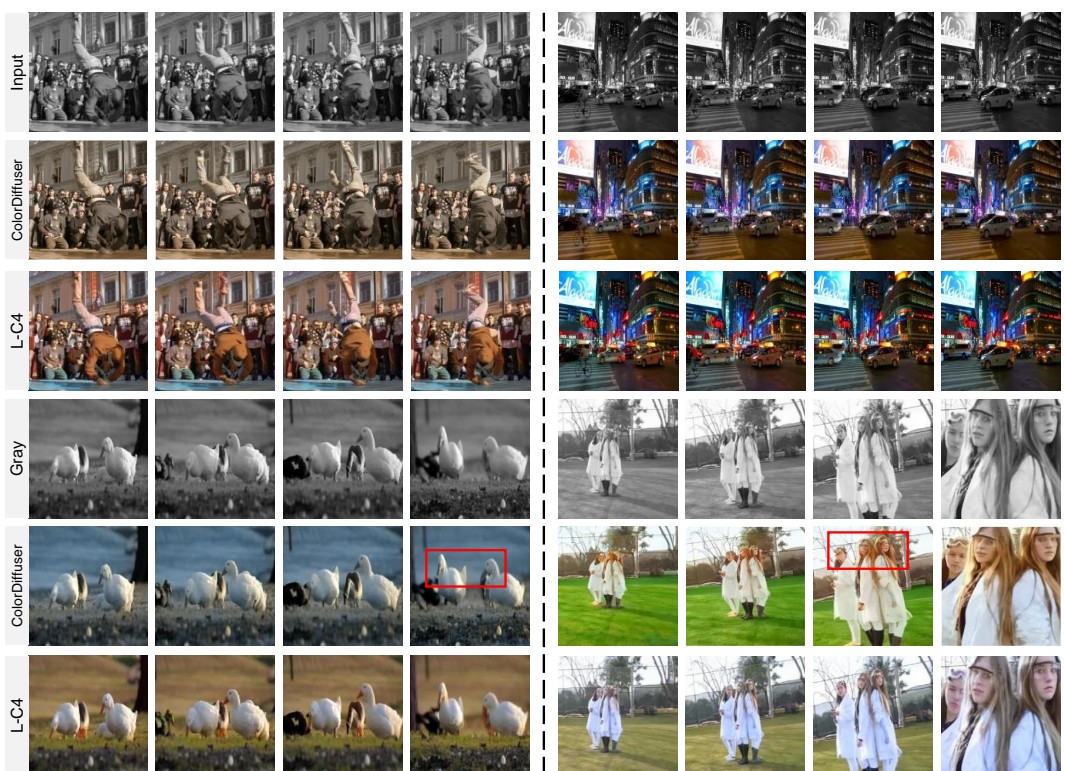

Figure 10: Additional visual quality comparison results with ColorDiffuser (Liu et al., 2023).

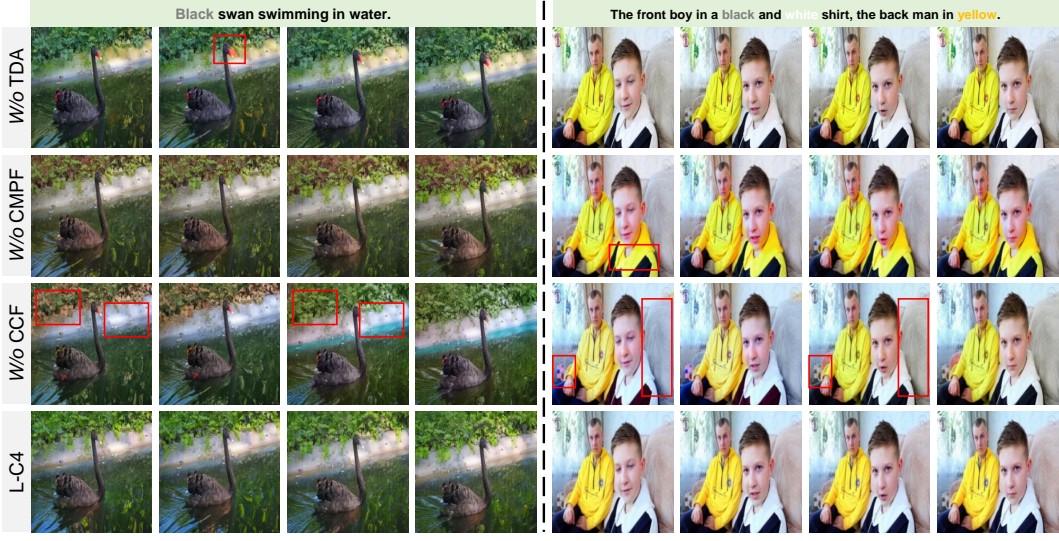

Figure 11: Additional visual quality comparison results with created baselines.

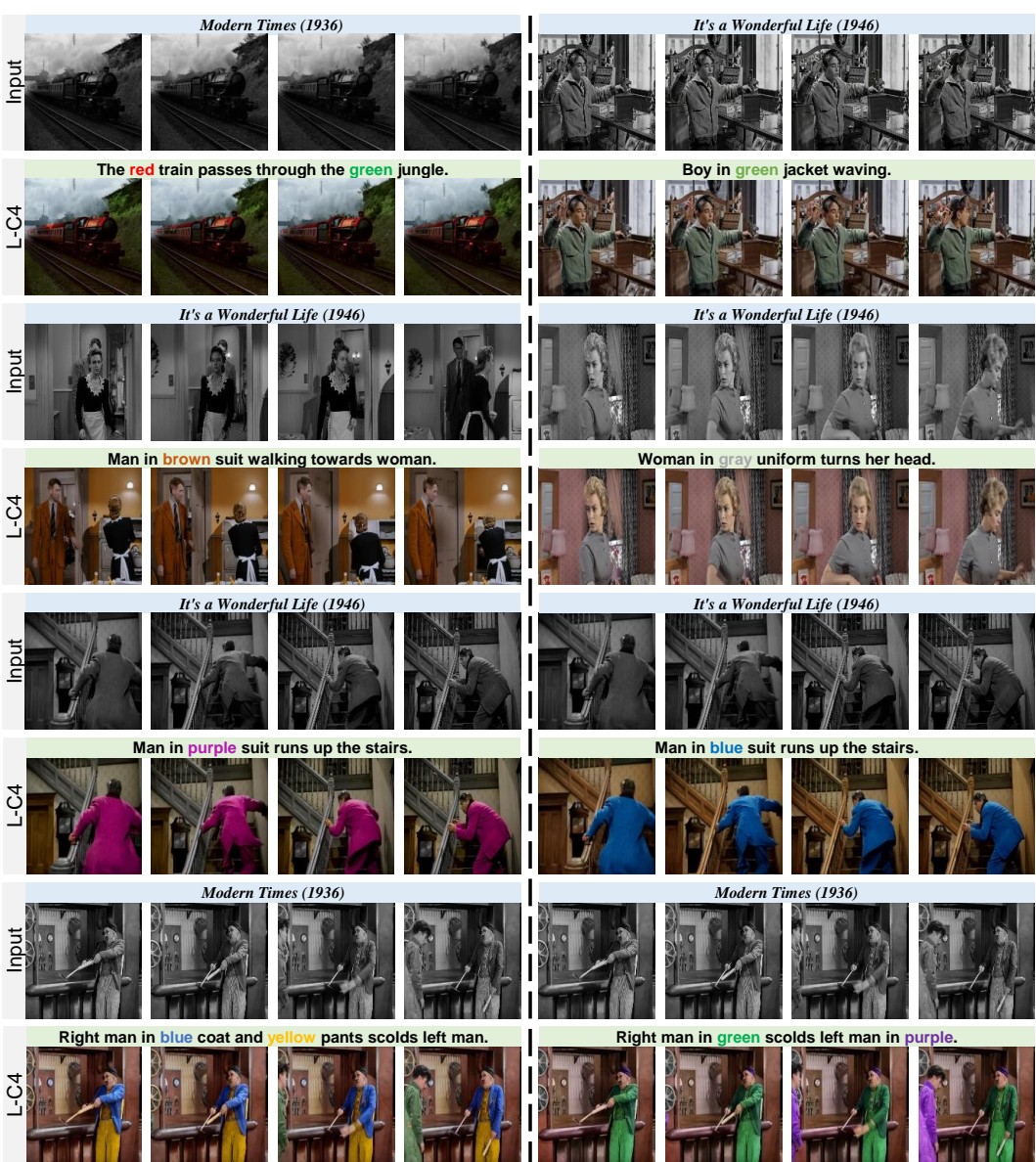

Figure 12: Application in old black-and-white film restoration.

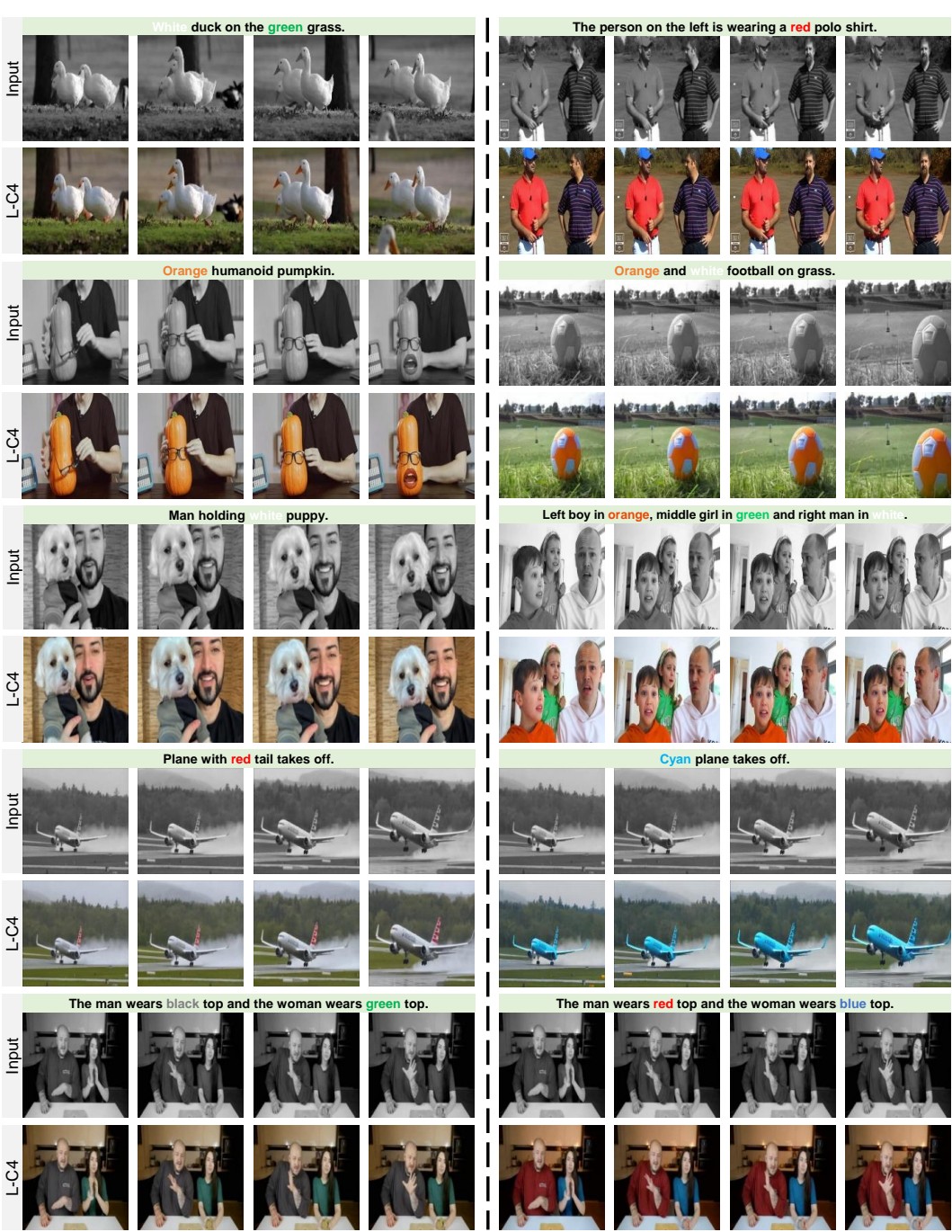

Figure 13: Application in real-world video restoration.

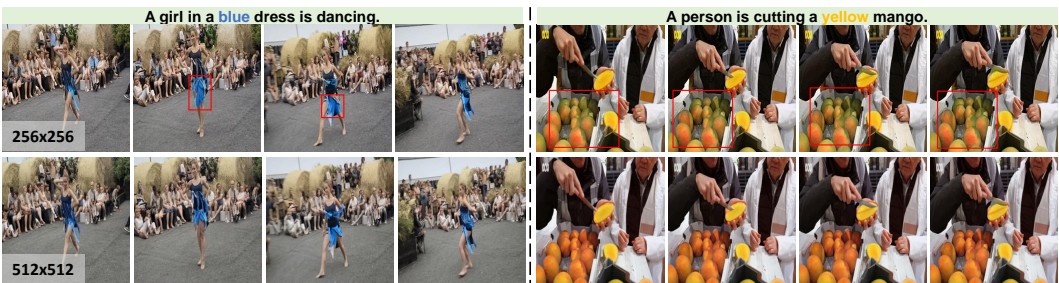

Figure 14: Colorization results are shown for L-C4 at a resolution of 256 pixels (top) and 512 pixels (bottom).

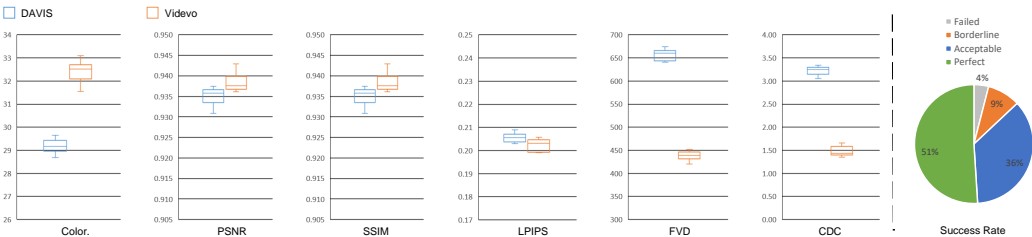

Figure 15: The box plots of quantitative metrics and the success rate from an additional user study.

Table 5: Additional quantitative comparison results with an automatic image colorization method (DDColor (Kang et al., 2023)) and a concurrent exemplar-based video colorization method (ColorMNet (Yang et al., 2024b)), using exemplars from the L-CAD (Chang et al., 2023a) for fairness.

| Method | DAVIS30 | | | | | | Videvo20 | | | | | |
|---|---|---|---|---|---|---|---|---|---|---|---|---|
| | Color. ↑ | PSNR ↑ | SSIM ↑ | LPIPS ↓ | FVD ↓ | CDC ↓ | Color. ↑ | PSNR ↑ | SSIM ↑ | LPIPS ↓ | FVD ↓ | CDC ↓ |
| DDColor | 21.67 | 22.66 | 0.881 | 0.223 | 1042.02 | 3.846 | 22.54 | 22.94 | 0.875 | 0.229 | 716.10 | 2.492 |
| DeepExemplar | 18.78 | 23.94 | 0.894 | 0.247 | 1168.38 | 4.126 | 22.02 | 23.07 | 0.901 | 0.248 | 747.06 | 1.941 |
| DeepRemaster | 14.52 | 23.87 | 0.907 | 0.239 | 916.39 | 5.389 | 14.40 | 23.12 | 0.893 | 0.219 | 796.50 | 3.642 |
| BiSTNet | 21.14 | 24.33 | 0.919 | 0.221 | 815.35 | 3.884 | 22.60 | 24.24 | 0.920 | 0.224 | 617.61 | 2.101 |
| ColorMNet | 19.35 | 25.20 | **0.935** | **0.206** | 1112.14 | 3.664 | 19.80 | **25.38** | **0.943** | 0.202 | 536.95 | 1.742 |
| Ours (L-C4) | **29.33** | **25.69** | 0.933 | 0.209 | **654.32** | **3.114** | **32.59** | 25.17 | 0.939 | **0.198** | **420.59** | **1.572** |

Table 6: Inference time for the comparison methods to colorize a 10-second video.

| DDColor | VCGAN | TCVC | DeepExemplar | DeepRemaster | BiSTNet | ColorMNet | L-CoIns | UniColor | L-CAD | Ours |
|---|---|---|---|---|---|---|---|---|---|---|
| 446s | 40s | 70s | 113s | 35s | 385s | 32s | 356s | 377s | 385s | 669s |

Table 7: Additional user study results with an expanded number of observers.

| Experiment | Dataset | VCGAN | TCVC | DeepRemaster | BiSTNet | UniColor | L-CAD | Ours |
|---|---|---|---|---|---|---|---|---|
| SCA | DAVIS30 | 5.8% | 8.2% | 10.4% | 14.2% | 12.6% | 13.2% | **35.6%** |
| | Videvo20 | 4.4% | 6.8% | 9.0% | 11.8% | 13.2% | 14.4% | **40.4%** |
| TCC | DAVIS30 | 7.8% | 10.6% | 10.8% | 11.6 % | 7.0% | 13.4% | **38.8%** |
| | Videvo20 | 15.2% | 16.8% | 5.6% | 11.4% | 6.8% | 7.6% | **36.6%** |

Table 8: Training datasets used for comparison methods.

| Method | DDColor | VCGAN | TCVC | DeepExemplar | DeepRemaster | BiSTNet |
|---|---|---|---|---|---|---|
| Dataset | ImageNet | ImageNet + DAVIS and Videvo | ImageNet + DAVIS and Videvo | Videvo stock and Hollywood2 | Subset of YouTube-8M | DAVIS and Videvo |
| Method | ColorMNet | L-CoIns | UniColor | L-CAD | Ours | - |
| Dataset | ImageNet + DAVIS and Videvo | Extended COCO-Stuff | ImageNet | Extended COCO-Stuff | Subset of InternVid | - |

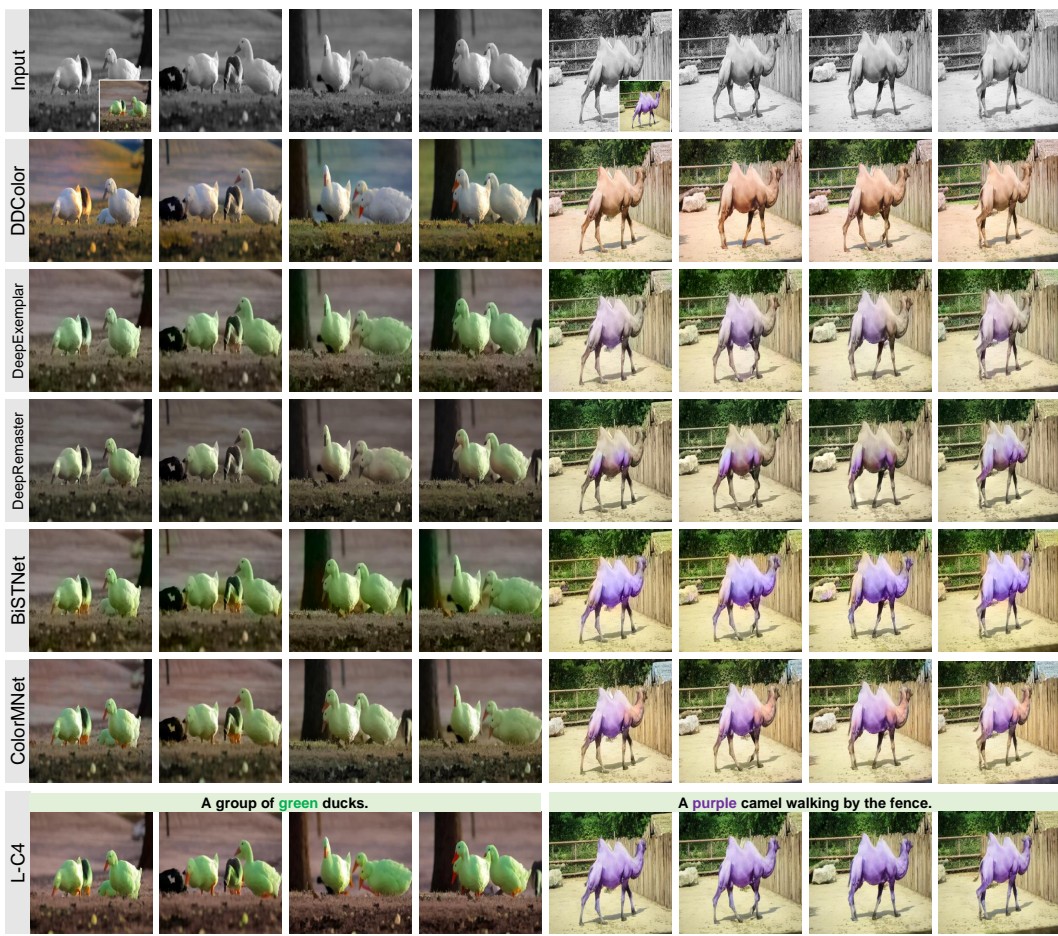

Figure 16: Additional qualitative comparisons with an automatic image colorization method (DD-Color (Kang et al., 2023)) and a concurrent exemplar-based video colorization method (ColorMNet (Yang et al., 2024b)).

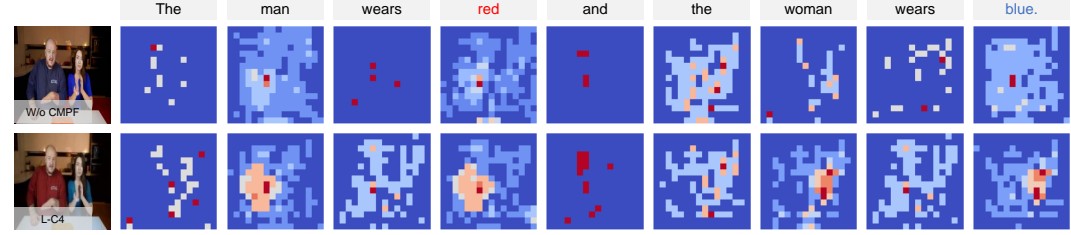

Figure 17: Visualization of attention maps from baselines shown without CMPF (top) and with CMPF (bottom).

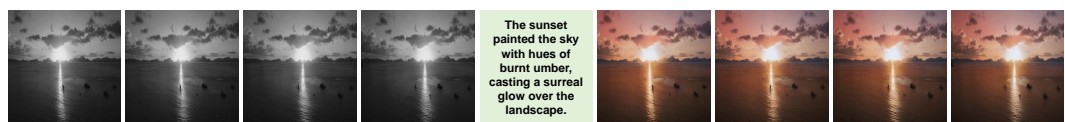

Figure 18: Colorization results with intricate prompts.

