# OpenReview forum: "L-C4: Language-Based Video Colorization for Creative and Consistent Colors"
_ICLR.cc/2025/Conference — Submitted to ICLR 2025_

### Official Review · Reviewer_3w9a · 2024-11-01

**Soundness:** 3
**Presentation:** 3
**Contribution:** 2
**Rating:** 3
**Confidence:** 4

**Summary:**

The paper proposes a language-based video colorization method (L-C4) to colorize monochrome videos based on the user-input instructions. The proposed method contains four main components, a cross-modality generative model [28] for language understanding and color modeling, a temporally deformable attention for inter-frame consistency, a cross-modality pre-fusion module for instance-aware text embedding generation, and a cross-clip fusion for long-term color consistency. The proposed method was evaluated on two existing datasets and show better results against several existing colorization methods.

**Strengths:**

1. The proposed method is technically sound.
2. Three main ablation results in Table 1 show the performance gains brought by the proposed techniques.
3. The paper is generally in good shape.
4. The results in Table 1 and the video are impressive.

**Weaknesses:**

1. Originality: The paper handles an existing task, video colorization, with an existing solution of using user-input languages as guidance [5,6,8,23,36]. The main difference is that while previous language-based methods are proposed for image colorization, the proposed method focuses on videos and considers modeling temporal consistency. However, the introduction does not discuss such differences and highlights the challenges and novelties. Comparing the language-based video colorization method to exemplar-based methods does not make sense, given the existence of language-based image colorization methods.

2. While the paper proposes a language-based video colorization method, the capability of understanding language and colors is from the existing image-based generation model [28]. Despite the proposed three methods (i.e., TDA, CMPF, and CCF), TDA and CCF focus on temporal coherency and CMPF is for text-instance alignment. There are no technical insights regarding color modeling from the colorization viewpoint.

3. The novelty of CMPF and CCF are not explained explicitly.

4. In fact, this paper is to extend a single image-based method [28] to handle the video colorization task. However, the paper does not provide a correct context (how existing methods handle temporal consistency and multi-modal alignment issues and what the true novelties of this work are) for evaluating its idea, and accordingly, the evaluations are not convincing (despite two experiments of L-CAD[4]+DF[20] and L-CAD[4]+temporal attention).

5. The authors claims that "Since the process of retrieving appropriate references can be elaborate, and the retrieved images are generally collected in the wild, exemplar-based video colorization methods may restrict the ability to assign creative colors to instances based on users’ imagination (Fig. 1 second row)". However, it is important to note that exemplar-based methods have the potential to incorporate language-based image colorization techniques, which can enable the assignment of creative colors. This raises the question of whether language-based video colorization methods are indeed necessary, or if the integration of language guidance into exemplar-based approaches could suffice to achieve the desired creative control over colorization. It would be beneficial for the authors to explore and discuss the limitations and advantages of both exemplar-based and language-based methods in the context of creative color assignment. A comparative analysis or experimental validation could strengthen their argument for the necessity of language-based video colorization methods over the adapted exemplar-based approaches.

Quality: (-) Most of the comparing methods are automatic and exemplar-based colorization methods, which are not close to the proposed method compared with those language-based image colorization methods [5,6,8,36,23] (the proposed method relies on the [28], which is also an image-based method). Meanwhile, there are also video diffusion models (as discussed in section 2.3) that study temporal coherency. Comparing the proposed method to (a combination of) language-based image colorization and video generation methods would be more convincing. (-) The proposed method lacks in-depth analysis and ablations, such as the sampling rate and hyperparameter alpha in TDA, the luminance encoder [6], and the hyperparameter N^f in CCF.

Clarity: (-) There are only some careless writings/typos: e.g., Temporal (Line 118) or Temporally (Line 54) Deformable Attention? Fig.12, last case (right): the colors should be red and blue. (-) The information about the training and evaluation datasets for existing methods and the proposed method is not clear, which may undermine the fairness of the comparisons.

**Questions:**

See weaknesses.

**Post rebuttal review:**

Thank you for your detailed response, which addressed some of my concerns. However, I still have several questions:

**W1:**

I am unclear on the necessity of employing **language-based video colorization**, given that diffusion models tend to be computationally intensive and slow, often requiring substantial GPU memory. Why not utilize **language-based image colorization** methods to generate high-quality exemplars and subsequently apply a lightweight exemplar-based video colorization approach? Such methods would be faster (32 seconds vs. 669 seconds) and yield superior results, as indicated by Tables 5 and 6.

**W2:**

The application of deformable convolution in video tasks has been extensively explored in prior research (e.g., Deformable ConvNets, and deformable attention in transformers). The proposed TDA module appears overly simplistic and lacks significant technical insights.

**W3:**

Regarding the Cross-Clip Fusion (CCF) strategy: Is it intended solely for sampling video clips? The authors state, "We introduce the Cross-Clip Fusion (CCF) to effectively extend the temporal interaction scope and maintain long-term color consistency when colorizing long videos." Have you tested this approach on long video sequences, such as those with a length of 500 frames? It seems that L-C4 may struggle to balance long-term performance with running time, as increasing ($N^f$) leads to a greater number of replicated features to fuse in CCF, resulting in longer inference time.

**W4:**

While the TDA and CCF modules focus on temporal coherence and CMPF is aimed at text-instance alignment, there appears to be a lack of technical insights related to color modeling from a colorization perspective.

**W5:**

This paper seems to extend a single image-based method to address the video colorization task. However, it does not adequately contextualize how existing methods manage temporal consistency and multi-modal alignment issues, nor does it clarify the true novelties of this work for evaluating its contributions.

**Quality-1:**

Table 1 indicates that on the DAVIS30 dataset, the configurations without TDA, CMPF, and CCF yield comparable CDC values. However, the integration of all three modules significantly improves the CDC by 0.479. Does this suggest that the video consistency is primarily derived from the network architecture rather than the proposed TDA? Additionally, Table 1 shows that on the Videvo20 dataset, the removal of CCF results in a degradation of CDC by approximately 0.8, indicating that CCF plays a critical role in maintaining consistency. Could you provide an explanation for this observation?

**Clarity:**

I appreciate that Figure 13 has already been updated for clarity.

**Final review:**

Thanks for the detailed response to my concerns

**W1:** I still have concerns regarding the evaluation results. The evaluations presented for both reference-based and non-reference-based metrics in Table 1 and Table 5 are confusing.
1. The Colorfulness score (Color. in Table 1) indicates the diversity of colors in the generated outputs, while PSNR, SSIM, and LPIPS measure fidelity to the ground truth. Typically, achieving high values in both metrics is challenging; for instance, a higher PSNR/SSIM/LPIPS often constrains the Colorfulness score due to the limited color range of the ground truth frames used for calculation. This phenomenon has been observed in numerous previous studies (e.g., Table 1 in the DeepExemplar paper, BiSTNet paper, and ColorMNet paper), where methods focusing on generating more colorful results, such as automatic image method, DDColor, achieve higher Colorfulness scores but relatively lower PSNR/SSIM/LPIPS. Conversely, exemplar-based methods prioritize fidelity, resulting in significantly higher PSNR/SSIM/LPIPS but lower Colorfulness scores (as demonstrated in Table 5 of this manuscript, comparing ColorMNet and BiSTNet). Have the authors assessed the Colorfulness score of the ground truth frames? It is unclear how a method can achieve both high PSNR and Colorfulness scores simultaneously.
2. Language-based methods assign colors based on descriptors (e.g., blue, white, green) but cannot specify exact hexadecimal values (e.g., #FFC0CB for pink). Unlike exemplar-based methods that utilize the first frame of the ground truth as an exemplar, language-based methods inevitably struggle with accurate color assignment and often cannot distinguish between similar colors (e.g., SeaGreen #2E8B57 and SpringGreen #3CB371). A slight error in color differentiation (such as mistaking SeaGreen for SpringGreen) can significantly impact PSNR, SSIM, or LPIPS outcomes. My primary concern is how L-C4, a language-based method, outperforms all exemplar-based methods (which use the first frame ground truth as an exemplar) in terms of PSNR, SSIM, and LPIPS.
3. The results for ColorMNet in Figure 16 are perplexing. The scenes depicted are relatively simple, characterized by straightforward objects and clean backgrounds. It is difficult to believe that ColorMNet fails to propagate color effectively, as shown with the camel in Figure 16, especially considering it utilizes DINOv2 to enhance feature extraction capabilities for more complex scenes. Furthermore, even the first frame—chosen as the exemplar—generated by ColorMNet exhibits significant color bleeding.

**W3:**
My concerns remain inadequately addressed. Specifically, does L-C4 require more processing time when $N^f$ increases? It seems that L-C4 may struggle to balance long-term performance with inference time, as increasing $N^f$ results in a greater number of replicated features to fuse in CCF, leading to extended inference duration.

**Quality-1:**
The ablation study in Table 1 does not convincingly validate the effectiveness of the proposed modules. The authors' responses have not adequately addressed my concerns.

Given the aforementioned issues and the lack of novelty highlighted by other reviewers, I will change my rating to 3.

---

### Official Review · Reviewer_RHjY · 2024-11-03

**Soundness:** 3
**Presentation:** 4
**Contribution:** 3
**Rating:** 8
**Confidence:** 4

**Summary:**

The paper introduces L-C4, a novel framework for language-based video colorization that guides the colorization process of monochrome videos using natural language descriptions, addressing the primary challenges faced in video colorization tasks: ambiguous color
assignment, limited creative imagination, and vulnerable color consistency. The proposed framework builds upon pre-trained cross-modal generative models to utilize their strong language understanding capabilities. To ensure accurate instance-level colorization, the paper proposes the Cross-Modality Pre-Fusion (CMPF) module to enhance text embeddings with instance-aware capability. Deformable Attention (TDA) block and Cross-Clip Fusion (CCF) are designed to ensure temporal color consistency across long sequences. The paper is well-written with clear motivation and rationales. The experiments are thorough, demonstrating their effectiveness.

**Strengths:**

- Each component is well-motivated, with clear explanations of how they address specific challenges in video colorization. The paper is well-written, with logical organization and flow that makes it easy to understand.

- The proposed Cross-Modality Pre-Fusion (CMPF) module enhances text embeddings with instance-aware capability, ensuring that the objects are assigned with desired colors, enabling creative and precise colorization given user input.

- The Temporally Deformable Attention (TDA) and Cross-Clip Fusion (CCF) effectively maintain color consistency across frames in a video. These components help prevent flickering and color shifts, leading to smoother and more visually appealing results.

- The experiments are thourough and well-designed, demonstrating superiority over state-of-the-art approaches.

**Weaknesses:**

- Regarding the generation of instance-aware embeddings, the paper does not fully explain or justify the creation of color-related masks. This might limit the model's applicability in complex cases, such as when dealing with intricate or lengthy textual descriptions.

- The paper lacks discussions on limitations.

- Some typos: Line 311, "cross-flip" should be "cross-clip".

**Questions:**

- What's the generation process of color-related masks? What if the text prompts are complicated or intricate? One example:
  "The sunset painted the sky with hues of burnt umber, casting a surreal glow over the landscape." Could the model handle these cases successfully, and to what extent will the quality and type of text prompts influence its performance?

- Temporally Deformable Attention (TDA) generates offset estimations for reference points. Would the performance get worse or better if we replace the offset estimation with an off-the-shelf model, such as an optical flow estimation model?

Thank the authors for the detailed explanation. My concerns have been fully addressed. Regarding the failure cases, I think that fine-grained color manipulation may be better controlled by other kinds of user interactions, such as reference images or strokes, given the high-level and semantic nature of text prompts. Overall, I think this is a paper that tackles text-guided video colorization problems with meaningful contributions and good demonstrations. I will raise my score.

---

### Official Review · Reviewer_jYuL · 2024-11-04

**Soundness:** 3
**Presentation:** 3
**Contribution:** 3
**Rating:** 6
**Confidence:** 4

**Summary:**

The paper introduces L-C4, a language-based video colorization framework designed to assign semantically accurate and creatively flexible colors to monochrome videos based on user-provided language descriptions. The proposed approach leverages a pre-trained cross-modality generative model and integrates three key components:

- **Cross-Modality Pre-Fusion (CMPF)**: Generates instance-aware text embeddings to facilitate precise color assignments based on language inputs.
- **Temporal Deformable Attention (TDA)**: Ensures inter-frame color consistency by dynamically capturing object features across frames.
- **Cross-Clip Fusion (CCF)**: Maintains long-term color consistency in extended video sequences by aggregating information across multiple clips.

Experimental evaluations on DAVIS, Videvo, and an additional VRIPT dataset demonstrate that L-C4 outperforms existing automatic, exemplar-based, and language-based image colorization methods across various quantitative and qualitative metrics. User studies further support the method's effectiveness in achieving temporally consistent and creatively satisfying colorizations.


**Post-rebuttal comment**: Having carefully considered the authors' rebuttal and fellow reviewers' perspectives, I maintain my recommendation of marginal accept. The authors have successfully clarified their contributions, which I appreciate. While there are some methodological limitations that remain - as noted by multiple reviewers including myself and acknowledged by the authors - I believe the work still makes a valuable contribution to the field. **These limitations, though noteworthy, do not overshadow the paper's positive aspects. Therefore, I support its borderline acceptance for publication.**

**Strengths:**

1. Innovative Integration of Components: The combination of CMPF, TDA, and CCF addresses key challenges in video colorization, such as instance-aware color assignments and temporal consistency.
2. Comprehensive Evaluation: The paper conducts extensive experiments, including quantitative metrics, qualitative analyses, user studies, and ablation studies, providing a well-rounded assessment of L-C4's performance.
3. Clear Motivation: The authors effectively identify the limitations of existing video colorization methods and articulate how L-C4 overcomes these issues through its novel framework.
4. State-of-the-Art Performance: L-C4 achieves superior results on multiple benchmarks, indicating its effectiveness in real-world applications like film restoration and artistic creation.

**Weaknesses:**

1. **Incremental Novelty**: While L-C4 combines existing techniques in a new way, components like deformable attention and cross-modality fusion are based on established methods. The paper would benefit from highlighting more unique innovations or providing deeper theoretical explanations for the observed performance improvements.

2. **Model Complexity and Efficiency**: Integrating TDA and CCF adds significant complexity to the pipeline, resulting in longer inference times and larger model sizes compared to baseline exemplar-based methods.

3. **Performance vs. Efficiency Trade-offs**: The paper does not discuss the balance between L-C4's performance gains and its increased computational costs, which is important for evaluating its practicality in real-world applications.

4. **Artifacts from Cross-Clip Fusion**: The cross-clip fusion approach may introduce artifacts or inconsistencies at clip boundaries, especially in videos with rapid motion or sudden scene changes.

5. **Fairness in Comparisons**: Despite the authors' efforts to ensure fair evaluations, comparing different methods remains challenging due to varying inputs and guiding signals.

**Questions:**

Please see the weakness section.

---

### Official Review · Reviewer_dnQZ · 2024-11-04

**Soundness:** 2
**Presentation:** 2
**Contribution:** 2
**Rating:** 3
**Confidence:** 5

**Summary:**

# Summary

The paper proposes a stable diffusion based video colorization framework. The paper leverages the temporal deformable attention for color consistency, cross-modality prefusion for instance-awareness, a skip window design for long-term temporal consistency. Experiments show the proposed method outperforms the selected baselines.

**Strengths:**

# Strengths

- The design of proposed submodules are reasonable and technically sound
- Extensive experiments demonstrate the effectiveness of the proposed framework

**Weaknesses:**

# Weaknesses

- The "Limited creative imagination" in L044 might not be a weakness of exemplar-based video colorization considering the user can use text-based image colorization methods to obtain exemplars
- It seems the paper overclaims its contribution in L091-092. It's unclear how reliable the proposed framework can "assign colors to each instance" considering the proposed framework still uses CLIP-based embeddings
- It seems that the design of LumEncoder is same as the previous works, but the paper does not cite them
- The paper does not mention how the exemplars were selected for the exemplar-based baselines. For L-C4 colorization with user prompts, the exemplar-based baselines might also need to adopt text-based colorization results as exemplar for fair comparisons
- Fig 4 and Fig 5 only contain one frame for each method, which seems inadequate to visualize the visual quality for video colorization. Visible color shifting artifacts can be observed in the supplementary video
- The paper does not mention the detailed evaluation settings: are prompts used in quantitative evaluations, if used, how they are obtained?
- Diffusion models can generate diverse results. The paper does not mention how results are selected for comparison and does not evaluate the success rate for each generation
- CDC metrics seem to have different scales with previous works
- Considering there is no ground truth for video colorization, using PSNR/SSIM for evaluation seems strange
- The paper does not compare with recent text-based video colorization methods like [1] and [2]
- The scale of user study seems relatively limited
- In Fig 7 right part, the boy's shirt in L-C4 seems suffering from color bleeding issue
- The paper does not mention runtime statistics, how long does the proposed framework take to colorize a video with 300 frames?

- [1] Towards Photorealistic Video Colorization via Gated Color-Guided Image Diffusion Models
- [2] Versatile Vision Foundation Model for Image and Video Colorization

**Questions:**

Please refer to the Weaknesses section

**EDIT:**

While the proposed framework appears reasonable, the paper has several major issues:

- overstating its contributions, even in the rebuttal
- evaluation results of baseline methods largely differ from those in the original papers. For example, prior to revision, unconditional colorization/using ground-truth frames as exemplars, the reported metrics of baseline methods in this paper seem significantly worse. Additionally, the rough metric-based ranking of all baseline methods also differs from what is reported in other papers.

As a result, I have decided to lower my rating to <5

---

### Official Review · Reviewer_FMrR · 2024-11-06

**Soundness:** 3
**Presentation:** 3
**Contribution:** 2
**Rating:** 5
**Confidence:** 5

**Summary:**

This paper proposes a language-based video colorization model and employs temporally deformable attention and fusion module to handle inter-frame color consistency and instance-aware colorization.

**Strengths:**

1. The paper is well-organized and well-written.
2. The design of the components in the proposed method generally makes sense.

**Weaknesses:**

Method Issues

1.	The proposed method can only process frames with fixed and low resolutions, specifically 256x256, making it incapable of handling high-resolution scenes. The results shown in the paper are also low-resolution, which might obscure some detailed issues. However, diffusion-based methods often struggle with details, so there are doubts about whether the proposed method can handle fine details effectively.

Performance Issues

1.	Lack of comparisons with state-of-the-art (SOTA) colorization methods. The comparisons shown in the paper omit more recent methods, such as ColorMNet[1] and DDColor[2].
2.	Some of the qualitative comparisons in the paper seem unfair and meaningless. For example, Figure 5 presents a comparison between the exemplar-based methods and the proposed text-based method under completely different conditions (e.g., the exemplar shows an orange fruit, while the text description specifies a pink fruit). Such comparisons lack significance, and the pink fruit in the figure also exhibits noticeable color overflow issues, where the pink in the highlight areas turns gray.
3.	Some of the videos in the demo still show color flickering and inconsistency issues, especially in small objects like shoes.
4.	Some of the displayed images still exhibit color bleeding issues.
5.	For diffusion-based methods, different seeds can lead to varying results. Does the proposed method perform consistently across different seeds? If not, what is the success rate of the proposed method?
6.	In video processing, processing speed is also essential. Please provide comparisons with the speeds of other video colorization methods.

[1] Yang, Yixin, et al. "ColorMNet: A Memory-based Deep Spatial-Temporal Feature Propagation Network for Video Colorization." ECCV, 2025.

[2] Kang, Xiaoyang, et al. "Ddcolor: Towards photo-realistic image colorization via dual decoders." ICCV. 2023.

**Questions:**

Please refer to weakness.

---

### Meta-Review · Area_Chair_dXCk · 2024-12-18

**Metareview:**

The submission introduces L-C4, a language-based video colorization method. The reviewers appreciated the novel integration of components addressing temporal consistency and creative flexibility. However, concerns were raised regarding computational efficiency, limited comparisons with recent methods, and overstated contributions. While some reviewers acknowledged the rebuttal and saw merit in the work, others remained unconvinced due to unresolved issues like performance consistency and evaluation clarity. The paper shows promise but requires refinements. The scores received post-rebuttal are 3, 3, 6, 8 and 5. Given the mixed reviews, the AC recommends a borderline reject.

**Additional Comments On Reviewer Discussion:**

Reviewers raised concerns about resolution limits, fairness of comparisons, computational efficiency, overstated claims, and novelty. Authors improved comparisons, retrained at higher resolution, clarified contributions, and addressed some efficiency concerns. Points on evaluation consistency and technical novelty remained unresolved for two reviewers (dnQZ, 3w9a). The mixed response led to a borderline reject recommendation.

---

### Decision · Program_Chairs · 2025-01-22

Reject